# Pregnant women's perceptions of the COVID-19 vaccine: A French survey

**Charles Egloff**[1,2,3☯]**, Camille Couffignal**[4,5☯]**, Anne Gael Cordier**[6,7]**, Philippe Deruelle**[8]**, Jeanne Sibiude**[1,2,3,4]**, Olivia Anselem**[9]**, Alexandra Benachi**[6,7]**, Dominique Luton**[2,4,10]**, Laurent Mandelbrot**[1,2,3,4]**, Christelle Vauloup-Fellous**[6,11,12]**, Alexandre J. Vivanti**[6,7]**, Olivier Picone**[1,2,3,4,11] *

1 Assistance Publique-Hôpitaux de Paris APHP, Nord, Service de Gynécologie Obstétrique, Hôpital Louis Mourier, Colombes, France, 2 Université de Paris, Paris, France, 3 INSERM, IAME, Paris, France, 4 FHU PREMA, Paris, France, 5 AP-HP, Hôpital Bichat, Clinical Research, Biostatistics and Epidemiology Department, Paris, France, 6 Université Paris-Saclay, INSERM U1193, Villejuif, France, 7 Department of Obstetrics and Gynecology, Antoine Beclere Hospital, Paris Saclay University, AP-HP, Clamart, France, 8 Pôle de gynécologie Obstétrique, Hôpitaux Universitaires de Strasbourg, Strasbourg, France, 9 Maternité Port-Royal, Hôpital Cochin, AP-HP, Centre-Université de Paris, Paris, France, 10 Service de gynécologie-obstétrique, FHU Prematurity, Bichat Hospital Assistance publique-Hôpitaux de Paris, Paris University, Paris, France, 11 Groupe de Recherche sur les Infections pendant la Grossesse (GRIG), Vélizy, France, 12 Laboratoire de Virologie, Hôpital Paul Brousse, AP-HP, Université Paris-Saclay, Villejuif, France

☯ These authors contributed equally to this work.
* Olivier.picone@aphp.fr

**Data Availability Statement:** All relevant data are within the manuscript and its Supporting information files.

## Abstract

### Introduction

Pregnant women are at increased risk for COVID-19, and COVID-19 vaccine is the most promising solution to overcome the current pandemic. This study was conducted to evaluate pregnant women's perceptions and acceptance of COVID-19 vaccination.

### Materials & methods

A cross-sectional study was conducted from February 18 to April 5 2021. An anonymous survey was distributed in 7 French obstetrics departments to all pregnant women before a prenatal visit. All pregnant women attending a follow-up consultation were asked to participate in the study. An anonymous web survey was available through a QR code and participants were asked whether or not they would agree to be vaccinated against SARS-CoV-2, and why. The questionnaire included questions on the patients' demographics and their knowledge of COVID-19 vaccines.

### Results

Of the 664 pregnant women who completed the questionnaire, 29.5% (95% CI 27.7; 31.3) indicated they would agree to be vaccinated against COVID-19. The main reason for not agreeing was being more afraid of potential side effects of the SARS-CoV-2 vaccine on the fetus than of COVID-19. Factors influencing acceptance of vaccination were: being slightly older, multiparity, having discussed it with a caregiver and acceptance of the influenza vaccine.

**Funding:** The authors received no specific funding for this work.

**Competing interests:** The authors have declared that no competing interests exist.

## Discussion

Nearly one-third of pregnant women in this population would be willing to be vaccinated. In addition to studies establishing fetal safety, public health agencies and healthcare professionals should provide accurate information about the safety of COVID-19 vaccines.

## Introduction

Since the beginning of the coronavirus disease 2019 (COVID-19) pandemic caused by severe acute respiratory syndrome coronavirus 2 (SARS-CoV-2), the number of confirmed cases and associated mortality and morbidity have increased rapidly (over 158 million confirmed cases, including over 3,000,000 deaths worldwide) [1]. Pregnant women are at increased risk for preterm births as well as ICU admission, mechanical ventilation and death [2–5].

Currently, no antiviral treatment has been effective in treating COVID-19 and vaccination is the most promising solution. Several international scientific societies strongly recommend that pregnant women have access to COVID-19 vaccines in all phases of future vaccine campaigns and that women and their healthcare professionals engage in shared decision-making regarding their receipt of the vaccine [6–8]. The French public health agency ("Direction Générale de la santé", DGS) recommends the administration of COVID-19 vaccines to all pregnant women from the second trimester of pregnancy [9]. The World Health Organization (WHO) states that pregnant women at high risk of exposure to SARS-CoV-2 (e.g. health workers) or who have comorbidities which add to their risk of severe disease may be vaccinated in consultation with their health care provider [10].

Vaccine safety scares, whether factual or fabricated, can erode confidence and reduce coverage. Acceptance of vaccination during pregnancy is likely to raise specific questions and concerns among pregnant women. The effects of vaccination on the placenta and fetus and physiological changes in pregnancy make pregnant women a specific population that may respond differently to vaccination [11–15]. These uncertainties must be considered in order to assess the benefit-risk balance and to make the most appropriate choice for pregnant patients at increased risk of severe COVID-19.

Beyond the logistics of implementation of mass vaccination and in order to best protect at-risk populations, adherence of the population to vaccination is a key point. Thus, we interviewed pregnant woman to evaluate their perceptions of the COVID-19 vaccine and their agreement to be vaccinated.

## Material and method

This cross-sectional study was conducted over a 6-week period from February 18 to April 5 2021 among pregnant women in 7 French obstetrics departments (Louis Mourier Hospital, Colombes; Béclère Hospital, Clamart; Bicêtre Hospital, Le Kremlin Bicêtre; Cochin Hospital, Paris; Bichat Hospital, Paris; Strasbourg Hospital: 2 centers). All pregnant women attending a follow-up consultation were asked to participate in the study. An anonymous web survey (Google Forms software®) was available through a QR code on consultation if the women agreed to participate. The questionnaire included questions on the patients' demographic information (age, geographic origins, level of education, profession), term of pregnancy, parity, comorbidity (obesity, high blood pressure, diabetes or other), personal history or close contact with COVID-19, knowledge of COVID-19 vaccination and flu vaccination habits for

current and previous pregnancies (S1 File). Participants were asked whether or not they would agree to be vaccinated against COVID-19, and why.

All statistical analyses were performed using R software (R Foundation for Statistical Computing, Vienna, Austria. http://www.r-project.org/) v. 4.0. For quantitative variables, we used descriptive statistics and the median and interquartile range. Discrete variables are presented as numbers and percentages. Missing data were not replaced. For the variables of interest, we used the 95% confidence interval. The collected data were compared considering the agreement or not to be vaccinated against COVID-19. A multivariate model was built to evaluate the factors associated with this agreement.

The study protocol was approved by the Institutional Review Board -IRB 00006477- of AP-HP.Nord, Paris University, AP-HP (N° CER-2021-67). Informed consent was obtained from patients in writing at the time of completing questionnaire. The collection of the data was fully anonymous: at the first data completion, the pregnant women were agreed that their data were using for this research. No minors were included.

## Results

A total of 664 pregnant women agreed to participate and completed the questionnaire.

### Demographics and characteristics

Demographic, pregnancy and and exposure to covid-19 or vaccination characteristics are described in Tables 1–4. Median age was 32 years (IQR [29–35]). Most patients were European (68.1%). Over half (258, 68.9%) had a bachelor's or higher university degree and 88.9% were employed, 13.1% of them as healthcare professionals. Half of the participants (50.9%) were in the third trimester of pregnancy and 33.9% had one or more comorbidities (principally diabetes, high blood pressure and obesity or overweight).

### Vaccination acceptance rate and influencing factors

In response to the question "Would you agree to be vaccinated against the COVID-19 virus?", 29.5% (95% CI 27.7;31.3) of the pregnant women answered "yes" (Table 5), and their main reasons for this acceptance were to protect themselves (84.7%), to protect relations (79.6%) or to protect the newborn (62.2%). Among those pregnant women who would not agree to be vaccinated, the main reason given was fear of side effects for their fetus (76.9%) and themselves (33.8%), rather than fear of COVID-19 infection. The two other major reasons for not to agreeing to COVID-19 vaccination were "Insufficient time for feedback on the side effects of the SARS-Cov-2 vaccine" (63.2%) and "Insufficient time for feedback on the effectiveness of the SARS-Cov-2 vaccine" (38.7%).

In bivariate analysis (Tables 1–4), factors still associated with vaccine acceptance rates were being older, being European, having a high educational level, being an executive, being multiparous, having discussed vaccination with a caregiver and being or wanting to be vaccinated against flu. Following adjustment for these seven factors, the pregnant women were more inclined to answer "Yes I agree to be vaccinated against SARS-CoV-2" if they were slightly older ($p < 0.001$), European ($p < 0.001$), already had at least one child ($p = 0.035$), had discussed it with a caregiver ($p < 0.001$) and were willing to be vaccinated against the flu ($p < 0.001$). In contrast, had a proven SARS-CoV-2 infection was not associated with acceptance rates ($p = 0,570$).

**Table 1. Pregnant women's sociodemographic characteristics and perceptions of the SARS-CoV-2 vaccine.**

| | All responders N = 664 | I agree to be vaccinated against SARS-CoV-2 | | p-value |
| --- | --- | --- | --- | --- |
| | | Yes N = 196 (29.5) | No N = 468 (70.5) | |
| **Age (years)** | 32 [29–35] | 33 [31–37] | 32 [29–35] | <0.001 |
| **Geographic origin** | | | | |
| Europe | 452 (68.1) | 158 (80.6) | 294 (62.8) | <0.001[a] |
| North Africa | 86 (13.0) | 15 (7.7) | 71 (15.2) | |
| Sub-Saharan Africa | 55 (8.3) | 5 (2.6) | 50 (10.7) | |
| Asia | 10 (1.5) | 1 (0.5) | 9 (1.9) | |
| Middle East | 6 (0.9) | 4 (2.0) | 2 (0.4) | |
| French Antilles and Guyana | 23 (3.5) | 1 (0.5) | 22 (4.7) | |
| Other | 32 (4.8) | 12 (6.1) | 20 (4.3) | |
| **Maternity hospital** (NA = 1) | | | | - |
| Bichat Hospital | 71 (10.7) | 22 (11.2) | 49 (10.5) | |
| Béclère Hospital | 174 (26.2) | 46 (23.5) | 128 (27.4) | |
| Louis Mourier Hospital | 151 (22.8) | 45 (23.0) | 106 (22.7) | |
| Bicêtre Hospital | 78 (11.8) | 13 (6.6) | 65 (13.9) | |
| Port-Royal Hospital | 95 (14.3) | 39 (19.9) | 56 (12.0) | |
| Strasbourg Hospital (center 1) | 51 (7.7) | 15 (8.2) | 35 (7.5) | |
| Strasbourg Hospital (center 2) | 43 (6.5) | 16 (7.7) | 28 (6.0) | |
| **Educational level** | | | | <0.001[b] |
| Middle or high school | 46 (7.1) | 9 (4.6) | 38 (8.1) | |
| High school diploma | 95 (14.4) | 11 (5.6) | 84 (18.0) | |
| Bachelor's degree | 52 (7.8) | 11 (5.6) | 41 (8.8) | |
| Master's degree | 222 (33.4) | 74 (42.9) | 138 (29.5) | |
| PhD | 184 (27.7) | 62 (31.6) | 122 (26.1) | |
| Other | 64 (9.6) | 19 (9.7) | 45 (9.6) | |
| **Profession** | | | | 0.004[c] |
| Employee | 183 (27.6) | 44 (22.5) | 139 (29.7) | |
| Executive | 241 (36.3) | 91 (46.4) | 150 (32.1) | |
| Healthcare professional | 87 (13.1) | 25 (12.8) | 62 (13.3) | |
| Unemployed | 37 (5.6) | 10 (5.1) | 27 (5.8) | |
| None | 38 (5.6) | 5 (2.6) | 32 (6.8) | |
| Other | 79 (11.9) | 21 (10.7) | 58 (12.4) | |

All data are presented by N (%) or median [IQR] p-value of Fisher's test for qualitative data or the Wilcoxon rank sum test for quantitative data.

[a] p-value of Fisher's test considering European origin versus other origin.

[b] p-value of Fisher's test considering four classes i.e. primary or secondary school; high-school diploma; bachelor's university degree; master's degree of PhD; other.

[c] p-value of Fisher's test considering four classes i.e. employee; executive; healthcare professional; unemployed or none.

NA: Not applicable/data missing.

## Discussion

Nearly one-third of patients would in theory be willing to be vaccinated. Patients who received information from a caregiver were more likely to accept vaccination. Accurate information from health care professionals should help to increase the rate of acceptance of COVID-19 vaccination.

**Table 2. Pregnancy characteristics and perceptions of the SARS-CoV-2 vaccine.**

| | All responders N = 664 | I agree to be vaccinated against SARS-CoV-2 | | p-value |
| --- | --- | --- | --- | --- |
| | | Yes N = 196 (29.5) | No N = 468 (70.5) | |
| **Comorbidity** | 0 [0–1] | 0 [0–1] | 0 [0–1] | |
| No comorbidity | 439 (66.1) | 132 (67.3) | 307 (65.6) | 0.719 |
| At least one comorbidity** | 225 (33.9) | 64 (32.7) | 161 (34.4) | |
| Diabetes | 52 (23.4) | 12 (18.8) | 40 (24.8) | |
| High blood pressure | 23 (10.4) | 9 (14.1) | 14 (8.7) | |
| Obesity or overweight | 69 (31.1) | 21 (32.8) | 48 (29.8) | |
| Other | 129 (58.1) | 38 (59.4) | 91 (56.5) | |
| **Weeks of amenorrhea** | | | | 0.497 |
| 12–18 | 114 (17.2) | 35 (17.9) | 79 (16.9) | |
| 19–24 | 111 (16.7) | 34 (17.4) | 77 (16.5) | |
| 25–30 | 101 (15.2) | 29 (14.8) | 72 (15.4) | |
| 31–36 | 192 (28.9) | 63 (32.1) | 129 (27.6) | |
| 37–42 | 146 (22.0) | 35 (17.9) | 111 (23.7) | |
| **Total pregnancies** | 1 [0–2] | 1 [0–2] | 1 [0–2] | 0.201 |
| **Total births** | 0 [0–1] | 1 [0–1] | 0 [0–1] | 0.032 |

All data are presented by N (%) or median [IQR] p-value of Fisher's test for qualitative data or the Wilcoxon rank sum test for quantitative data.

**pregnant women with at least one comorbidity, median [IQR] (range) 1 [1–1] (1–3).

NA: Not applicable/data missing.

## Acceptance rates among pregnant women compared to the general population

As expected, our results indicated a lower rate of acceptance of COVID-19 vaccination among pregnant women, i.e. 29.5%, than in the general population. In a recent meta-analysis,

**Table 3. Exposure to COVID-19 and perceptions of the SARS-CoV-2 vaccine.**

| | All responders N = 664 | I agree to be vaccinated against SARS-CoV-2 | | p-value |
| --- | --- | --- | --- | --- |
| | | Yes N = 196 (29.5) | No N = 468 (70.5) | |
| **Have you had a proven SARS-CoV-2 infection?** | | | | 0.570 |
| Yes | 66 (9.9) | 17 (8.7) | 49 (10.5) | |
| No | 598 (90.1) | 179 (91.3) | 419 (89.5) | |
| **If so, were you hospitalized?** (Na = 1) | | | | - |
| Yes | 2 (3.1) | 1 (5.9) | 1 (2.1) | |
| No | 63 (96.9) | 16 (94.1) | 47 (98.0) | |
| **Have you had close contact (family or friend) with someone who had a SARS-CoV-2 infection?** | | | | 1 |
| Yes | 180 (27.1) | 53 (27.0) | 127 (27.1) | |
| No | 484 (72.9) | 143 (73.0) | 341 (72.9) | |
| **If so, was the contact hospitalized?** (NA = 2) | | | | - |
| Yes | 14 (7.9) | 4 (7.7) | 10 (7.9) | |
| No | 164 (92.1) | 48 (92.3) | 116 (92.1) | |

All data are presented by N (%) or median [IQR] p-value of Fisher's test for qualitative data or the Wilcoxon rank sum test for quantitative data.

NA: Not applicable/data missing.

**Table 4. Exposure and knowledge of vaccination and perceptions of the SARS-CoV-2 vaccine.**

| | | I agree to be vaccinated against SARS-CoV-2 | | |
|---|---|---|---|---|
| | All responders N = 664 | Yes N = 196 (29.5) | No N = 468 (70.5) | p-value |
| **Do you know that a SARS-CoV-2 vaccine exists?** | | | | 0.636 |
| Yes | 643 (96.8) | 191 (97.4) | 452 (96.6) | |
| No | 21 (3.2) | 5 (2.6) | 16 (3.4) | |
| **Have you ever discussed vaccination against SARS-CoV-2 with a healthcare professional?** | | | | <0.001 |
| Yes | 191 (28.8) | 78 (39.8) | 113 (24.2) | |
| No | 473 (71.2) | 118 (60.2) | 355 (75.8) | |
| **If so, what type?** | | | | - |
| Obstetrician-gynecologist | 84 (44.0) | 38 (48.7) | 46 (40.7) | |
| Midwife | 36 (18.8) | 17 (21.8) | 19 (16.8) | |
| General practitioner | 35 (18.3) | 10 (12.8) | 25 (22.1) | |
| Other | 36 (18.8) | 13 (16.7) | 23 (20.4) | |
| **Have you been vaccinated against the flu this year or in previous years in connection with a previous pregnancy?** | | | | <0.001 |
| Yes | 285 (42.9) | 116 (59.2) | 169 (36.1) | |
| No | 377 (56.8) | 80 (40.8) | 297 (63.5) | |
| First pregnancy | 2 (0.3) | 0 (0.0) | 2 (0.4) | |
| **If not, why?** (Na = 29) | | | | |
| Not wanted | 229 (65.8) | 20 (27.0) | 209 (76.3) | |
| Not offered | 119 (34.2) | 54 (73.0) | 65 (23.7) | |

All data are presented by N (%) or median [IQR] p-value of Fisher's test for qualitative data or the Wilcoxon rank sum test for quantitative data.

**pregnant women with at least one comorbidity, median [IQR] (range) 1 [1–1] (1–3).

NA: Not applicable/data missing.

**Table 5. Pregnant women's acceptance of SARS-CoV-2 vaccination.**

| | All responders N = 664 |
|---|---|
| **Would you agree to be vaccinated against SARS-CoV-2?** | |
| Yes | 196 (29.5) |
| No | 468 (70.5) |
| **If so, why?** | |
| To protect me | 166 (84.7) |
| To protect my relations | 156 (79.6) |
| To reduce the risk of COVID-19 infection in my future child | 122 (62.2) |
| **If not, why?** | |
| **I am more afraid of the side effects of the SARS-CoV-2 vaccine on my fetus than of COVID** | **360 (76.9)** |
| I am more afraid of the side effects of the SARS-CoV-2 vaccine on me than of COVID | 158 (33.8) |
| Insufficient time for feedback on the side effects of the SARS-CoV-2 vaccine | 296 (63.2) |
| It depends on the type of vaccine | 28 (6.0) |
| Insufficient time for feedback on the effectiveness of the SARS-CoV-2 vaccine | 181 (38.7) |
| Other | |
| *Not necessarily, considering the health status* | *4 (0.8)* |
| *No specific study on pregnant woman* | *19 (4.1)* |
| *After delivery or breastfeeding* | *15 (3.2)* |
| *Concomitant disease (chronic disease, factor V mutations, allergy...)* | *12 (2.6)* |

* first choice for this multiple-choice question.

Robinson et al. found across 13 countries that the pooled proportion reporting intention to vaccinate was 0.729 (05% CI 0.666–0.996) vs 0.143 (95% CI 0.114–0.179) for patients who would refuse a vaccine and 0.221 (95% CI 0.178–0.271) for those who were unsure [16]. Interestingly, COVID-19 vaccination acceptance seems to have declined over time since the beginning of the pandemic. These results are similar to those of a French national survey that found that COVID-19 vaccine refusal has steadily increased between May 2020 and October 2020 [17].

## Acceptance rate compared to other pregnant populations

Usually, pregnancy decreases the acceptance rate for vaccination. A review published in 2015 found that the main barriers to vaccination acceptance in the pregnant population were related to vaccine safety, belief that vaccine is not needed or effective, not recommended by healthcare workers and low knowledge about vaccines [18]. In a recent study, Skjefte et al. evaluated COVID-19 vaccine acceptance rate among pregnant women. Using an online anonymous survey across 16 countries and 5294 pregnant women, COVID-19 vaccine acceptance level varied substantially by country (80% for pregnant women in Mexico and India; below 45% for the USA, Australia and Russia) [19]. No French women were represented, but in other European countries (i.e. Italy, Spain, United Kingdom), COVID-19 acceptance rate was approximately 45%, which is higher than what we found in this study. This can be explained by the fact that two of the strongest predictors of non-acceptance of COVID-19 vaccines were lower educational level and lower income. In our cohort, the survey was accessed using a QR code, and this may be a limitation of our study. Furthermore, the survey by Skjefte et al. was conducted before (i.e. between October 28 and November 18, 2020) the Federal Drug Administration (FDA) issued the first Emergency Use Authorization (EUA) for Pfizer-BioNTech's mRNA COVID-19 vaccine, and assuming that the vaccine was safe and free. These authors' data, like ours, indicated that the main reasons pregnant women declined COVID-19 vaccination during pregnancy were that they did not want to expose their fetus to any possible harmful side effects and would like to see more safety and effectiveness data among pregnant women. Skjefte et al. found that slightly less than half of pregnant women did not want vaccination because they "were concerned that approval of the vaccine would be rushed for political reasons". This factor was not addressed in our survey, although it is closely related to the acceptance rate, but probably varies over time and is difficult to assess.

In their multinational, cross-sectional, web-based study conducted in six European countries at the beginning of the COVID-19 pandemic (between April and July 2020), Ceulemans et al. found that 61% of the pregnant respondents would agree to be vaccinated if a vaccine was available [20]. If Belgium is excluded from the sample (because the data were collected earlier, i.e. between April and May vs June and July for the others), the proportion of pregnant women who would like to be vaccinated dropped to 49%. Even though this population is similar to ours (i.e. European), the study was conducted at the beginning of the epidemic, which makes comparisons difficult, but confirms the decrease in acceptance rate over time seen previously.

Finally, Carbone et al, using a methodology similar to ours, found an acceptance rate of 28.2% among 142 Italian patients in their study conducted in January 2021 [21].

## Reasons for acceptance of vaccination and prognostic factors

The reasons associated with refusal of COVID-19 vaccination were that the women were more afraid of the side effects on their fetus (76.9%) and themselves (33.8%) than of COVID-19 infection. The two other major reasons for not agreeing to COVID-19 vaccination were

"Insufficient time for feedback on the side effects of the SARS-Cov-2 vaccine" (63.2%) and "Insufficient time for feedback on the effectiveness of the SARS-Cov-2 vaccine" (38.7%). These reasons are also the main ones found in the other studies (19–21).

Prognostic factors associated with acceptance of COVID-19 vaccination were being slightly older, being European, already having at least one child, discussing it with a caregiver and being or wanting to be vaccinated against the flu. Carbone et al. found no statistically significant differences, maybe due to lack of power given the small number of patients. In the two large online surveys, low educational level was associated with non-acceptance of COVID-19 vaccination. Skjefte et al. found that the strongest predictors of acceptance were older age, higher income, and belief in the importance of vaccines. While in our study multiparity was associated with greater acceptance of COVID-19 vaccination, Ceulemans et al. and Carbone et al. found the opposite result. In their studies, information on livebirths and history of miscarriage is not specified and may influence the participants' responses. Our results do not show a significant difference between history of exposure to SARS-Cov-2 and acceptance rate to be vaccinated. This may be related to the fact that patients with a previous SARS-CoV-2 infection believe they are still protected. Thus, patients with severe disease requiring hospitalization appear to be more likely to be vaccinated (5,9% vs 2,1%), although due to the small number of hospitalized patients, statistical analysis could not be performed.

Interestingly, our study is the only one to focus on information given by a caregiver and we found that pregnant women who discussed vaccination with their caregiver were more likely to be willing to get the vaccine. This finding highlights the importance of the information campaign that must be conducted among patients by health professionals. It is also reasonable to think that the rate of COVID-19 vaccination acceptance has probably increased since the collection of the questionnaires. Our study was conducted in France, after the Collège National de Gynécologie et Obstétrique Français (CNGOF, the French College of Obstetricians and Gynecologists) issued its recommendations and before the Direction Générale de la Santé (DGS, a department of the French Ministry of Solidarity and Health) recommended vaccination for pregnant women. While the CNGOF has only an advisory role, the DGS is responsible for preparing public health policy and contributing to its implementation. At the time of our survey, access to vaccination for all pregnant women was not yet established, and this probably limited the acceptance rate.

During this pandemic, the opinions of different organizations and experts could sometimes differ and be contradictory, making it difficult for practitioners to advise patients. Thus, it is the duty of the main learned societies to issue recommendations, making it possible to guide health professionals in order to have a univocal and clear discourse for patients.

## Knowledge about the efficacy and safety of COVID-19 vaccination during pregnancy

Despite their higher risk, pregnant women were excluded from initial vaccination clinical trials. The limited data from animal studies and phase II/III clinical trials suggest that the vaccines do not harm embryonic development in animals if given during pregnancy and no side effects was found in 53 pregnancies that occurred across the trials [22]. The first randomized clinical trial evaluating the safety, tolerability, and immunogenicity of a COVID-19 mRNA vaccine on pregnant woman began in February 2021 and should end in June 2022 *(Pfizer/BioNTech, ClinicalTrials.gov. Identifier: NCT04754594)*. After our study was completed, Shimabukuro et al. published safety data from a voluntary post-vaccination follow-up registry [23]. No obvious safety signals were highlighted. The incidences of adverse pregnancy and neonatal outcomes (i.e. fetal loss, preterm birth, small size for gestational age, congenital

anomalies, and neonatal death) among vaccinated pregnant women were similar to the incidences reported in pregnant populations studied before the COVID-19 pandemic. Among pregnant women who reported congenital anomalies, none had received a COVID-19 vaccine in the first trimester or periconceptional period, and no specific pattern of congenital anomalies was observed. However, data from completed pregnancies were mainly limited to patients vaccinated in the third trimester of pregnancy and further study is needed of outcomes (especially miscarriage and congenital anomalies) after vaccination in the first trimester. In addition, studies with longer pediatric follow-up should focus on the psychomotor outcome of children in the next few years. Furthermore, COVID-19 mRNA vaccine administration during pregnancy is responsible for a good immune response with significant antibody levels similar to those in non-pregnant women. Among the 33 patients who delivered in the studies by Gray et al. and Rottenstreich et al., anti-SARS-CoV-2 antibodies were present in all cord blood samples [24,25].

## Limitations of our study

The main limitation of our study is response bias, which may limit the representativeness of our population. We analyzed the sociodemographic results of our population in relation to the results of the French National Perinatal Surveys conducted from 1995 to 2016 [26]. These large surveys carried out in France for one week every 5 years describe the health status of mothers and newborns, their characteristics, and medical practices during pregnancy and at the time of delivery. In our population, 68% of patients were from Europe, while in the 2016 French National Perinatal Survey 85.6% of the pregnant woman were French. However, the question asked in our questionnaire was: "Where are you from?" and there was no specific question on nationality, unlike the French Perinatal Survey, which makes comparisons difficult. The level of education of our population was similar to that of the general population: 21.5% of our population had not continued their studies after a high-school diploma, vs. 22.9% in the French National Perinatal Survey. Furthermore, the obstetrics departments that participated in our study are tertiary care centers that manage patients with comorbidities. In our cohort, 10.4% of pregnant women reported high blood pressure and 23.4% reported diabetes, compared with the rates of 4.3% and 10.8%, respectively, in the population of pregnant women in the 2016 French National Perinatal Survey.

An other limitation is that our study was conducted between February 18 to April 5 2021, during a turbulent period regarding the media coverage of vaccination in the French and European country. In particular, the Astra Zeneca vaccine, although never recommended for pregnant women, has been the source of much concern following reports of rare adverse events. Survey responses could be influenced by such this labile context.

## Conclusion

Nearly one-third (29.5%) of our patients would be willing to be vaccinated before consultation. The main reason for non-acceptance was the fear of side effects in the fetus. Prognostic factors associated with acceptance were being slightly older, European, already having at least one child, discussing it with a caregiver and being or wanting to be vaccinated against the flu. In addition to studies allowing reassurance of pregnant women regarding fetal risk, public health agencies and healthcare professionals need to provide accurate information about the safety of COVID-19 vaccines.

## Supporting information

**S1 File.**
(DOCX)

## Author Contributions

**Conceptualization:** Anne Gael Cordier, Philippe Deruelle, Christelle Vauloup-Fellous, Olivier Picone.

**Data curation:** Charles Egloff.

**Formal analysis:** Charles Egloff, Camille Couffignal, Olivier Picone.

**Investigation:** Olivier Picone.

**Methodology:** Camille Couffignal, Olivier Picone.

**Project administration:** Camille Couffignal, Olivier Picone.

**Supervision:** Olivier Picone.

**Validation:** Jeanne Sibiude, Olivia Anselem, Alexandra Benachi, Dominique Luton, Laurent Mandelbrot, Alexandre J. Vivanti, Olivier Picone.

**Writing – original draft:** Charles Egloff, Camille Couffignal.

**Writing – review & editing:** Anne Gael Cordier, Philippe Deruelle, Alexandre J. Vivanti, Olivier Picone.

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
