## [Decision Letter · Decision Letter 0]

29 Oct 2021

PONE-D-21-16178Pregnant women’s perceptions of the COVID-19 vaccine: a French national survey

PLOS ONE

Dear Dr. Picone,

Thank you for submitting your manuscript to PLOS ONE. After careful consideration, we feel that it has merit but does not fully meet PLOS ONE’s publication criteria as it currently stands. Therefore, we invite you to submit a revised version of the manuscript that addresses the points raised during the review process.

Kindly provide a stronger rationale for doing this study and the future implications of the findings. Also, provide an explanation of the key variables used in the study and how they are measured.

We look forward to receiving your revised manuscript.

Kind regards,

Russell Kabir, PhD

Academic Editor

PLOS ONE

Journal Requirements:

4. Please include additional information regarding the survey or questionnaire used in the study and ensure that you have provided sufficient details that others could replicate the analyses. For instance, if you developed a questionnaire as part of this study and it is not under a copyright more restrictive than CC-BY, please include a copy, in both the original language and English, as Supporting Information.

"the funders had no role in study design, data collection and analysis, decision to publish, or preparation of the manuscript."

7. Please include a copy of Table S1 which you refer to in your text on page 4.

Reviewers' comments:

Reviewer's Responses to Questions

**Comments to the Author**

1. Is the manuscript technically sound, and do the data support the conclusions?

Reviewer #1: Yes

Reviewer #2: Partly

2. Has the statistical analysis been performed appropriately and rigorously? 

Reviewer #1: Yes

Reviewer #2: Yes

3. Have the authors made all data underlying the findings in their manuscript fully available?

Reviewer #1: Yes

Reviewer #2: Yes

4. Is the manuscript presented in an intelligible fashion and written in standard English?

Reviewer #1: Yes

Reviewer #2: Yes

5. Review Comments to the Author

Reviewer #1: Table 1 should be break down into 4 separate tables Like

table 1: Pregnant women's sociodemographic characteristics and perceptions of the SARS-CoV-2 vaccine

table 2: Pregnancy characteristics and perceptions of the SARS-CoV-2 vaccine

table 3: Exposure to COVID-19 and perceptions of the SARS-CoV-2 vaccine

table 4: Exposure and knowledge of vaccination and perceptions of the SARS-CoV-2 vaccine

Relationship between previous exposure to COVID-19 and Vaccine acceptance should be highlighted both in results and discussion section

Reviewer #2: Thank you for this paper employing a cross-sectional survey on attitudes to vaccination for pregnant women. On the whole I am happy with the methodology used in pressing circumstances. The results and their implications concerning the reasons for being reticent about vaccination, risk for the fetus, protection of oneself and population are well reported and discussed from a clinical perspective and how they might affect willingness to get vaccinated. So no especial quarrels concerning the heart of the paper. I would however like to discuss two matters which could be further developed.

Firstly, I feel that for readers to appreciate this study it is important to add further details about the context in which the study was carried out. As you are well aware the period in which the study took place was to say the least fairly turbulent with regard to how vaccination was being reported in the French and European media (not to mention social media) and indeed also on a more specialist front with respect to advice being offered by different agencies and expert bodies. AS you rightly point out this differed in emphasis concerning vaccination for pregnant women.

It is well-accepted that cross-sectional studies are undertaken at a point of time or over a short period of time. Nevertheless although single individuals are being questioned with regard to their attitudes at a single point of time, the context in which individual attitudes are being investigated was in this particular case rather labile. Notably with regard to Astra zenica. Even if this was not the vaccine of choice against Covid one cannot help but feel that it is difficult not to take into account in your discussion of results and possible limitations

Astra zenica was at the centre of a media and Health agency storm in early March. After having been launched on 25Th February 2021, following reports of rare adverse effects, it was suspended for use on 15/03 2021 and then reauthorized on 19th March. Could survey responses be influenced by such a labile context?

Further more advice being offered as the authors duly point out varied between agencies and scientific bodies at the time. Thus advice from different agencies differed at least in strength of recommendation for pregnant women to get vaccinated. This also may have fueled the prevailing uncertainty. and although perhaps not as media worthy, was as you are well-aware aired in the medical press accessible on line.

It is easy to imagine that this turbulent context could have had an influence on attitudes, even if subsequently Astra Zenica was not recommended for administration to pregnant women. The vaccines of choice being Moderna and BioNtech Pfizer.

Evoking context raises a big issue in social psychology . ( See Schwarz,2007) Should attitudes be considered as being dispositional i.e. people have attitudes and they remain rather stable. An observers perspective. This downplays context. Or rather as construals, Evaluative judgements constructed in situations by taking into account the context and possible circulating information An actor’s perspective This highlights how attitudes can be context sensitive. On could also suggest that a changing epidemiological context may affect how people feel about Covid reducing or augmenting perceived risk.

Could attitudes be affected or not affected, by such events and differences in advice? How could this affect the cross-sectional study even if the period over which the study was conducted was relatively short? At the time vaccination was not yet available to all of the pregnant women Furthermore as far as I can ascertain attitudes to different vaccines understandably were not taken into consideration within the questionnaire.

These matters I feel merit discussion in the paper or at least cited as a possible limitation.

A second difficult issue concerns clinical advice and its relationship to research evidence. This deserves reflection. It makes good sense to offer advice to any person seeking vaccination, all the more so in the case of pregnant women. Furthermore as the authors point out probably the best people to give health information are the health professionals caring for the pregnant women themselves since trust in a clinical relationship is paramount.

Nevertheless when advice by leading bodies and experts (including members of the research team who took an active role in the debate ) is differing how can such uncertainty be tackled in clinical practice? At what point does research evidence cease to be part of ‘science in the making’ and become ‘ready made science’ ( cf Science in Action, Latour)

6. PLOS authors have the option to publish the peer review history of their article (what does this mean?). If published, this will include your full peer review and any attached files.

Reviewer #1: No

Reviewer #2: No

---

## [Author Response · Author response to Decision Letter 0]

23 Nov 2021

We thank the reviewers for the interesting comment that we took into account in the new version of the manuscript. All the detailled answers are presented in a specific file.

---

## [Decision Letter · Decision Letter 1]

21 Jan 2022

Pregnant women’s perceptions of the COVID-19 vaccine: a French survey

PONE-D-21-16178R1

Dear Dr. Olivier Picone,

We’re pleased to inform you that your manuscript has been judged scientifically suitable for publication and will be formally accepted for publication once it meets all outstanding technical requirements.

Kind regards,

Sharon Mary Brownie

Academic Editor

PLOS ONE

Reviewers' comments:

Reviewer's Responses to Questions

**Comments to the Author**

1. If the authors have adequately addressed your comments raised in a previous round of review and you feel that this manuscript is now acceptable for publication, you may indicate that here to bypass the “Comments to the Author” section, enter your conflict of interest statement in the “Confidential to Editor” section, and submit your "Accept" recommendation.

Reviewer #1: All comments have been addressed

Reviewer #2: All comments have been addressed

2. Is the manuscript technically sound, and do the data support the conclusions?

Reviewer #1: Yes

Reviewer #2: (No Response)

3. Has the statistical analysis been performed appropriately and rigorously? 

Reviewer #1: Yes

Reviewer #2: (No Response)

4. Have the authors made all data underlying the findings in their manuscript fully available?

Reviewer #1: Yes

Reviewer #2: (No Response)

5. Is the manuscript presented in an intelligible fashion and written in standard English?

Reviewer #1: Yes

Reviewer #2: (No Response)

6. Review Comments to the Author

Reviewer #1: (No Response)

Reviewer #2: (No Response)

7. PLOS authors have the option to publish the peer review history of their article (what does this mean?). If published, this will include your full peer review and any attached files.

Reviewer #1: No

Reviewer #2: **Yes: **William Sherlaw

---

## [Editor Report · Acceptance letter]

28 Jan 2022

PONE-D-21-16178R1 

Pregnant women’s perceptions of the COVID-19 vaccine: a French survey 

Dear Dr. Picone:

I'm pleased to inform you that your manuscript has been deemed suitable for publication in PLOS ONE. Congratulations! Your manuscript is now with our production department. 

Kind regards, 

on behalf of

Professor Sharon Mary Brownie 

Academic Editor

PLOS ONE